# Covalent Immobilization of Antibodies through Tetrazine-TCO Reaction to Improve Sensitivity of ELISA Technique

**DOI:** 10.3390/bios11120524

**Published:** 2021-12-20

**Authors:** Tania García-Maceira, Fé I. García-Maceira, José A. González-Reyes, Luis A. Torres-Sánchez, Ana Belén Aragón-Gómez, María Eugenia García-Rubiño, Elier Paz-Rojas

**Affiliations:** 1Canvax Biotech, Parque Científico y Tecnológico Rabanales 21, c/Astrónoma Cecilia Payne s/n, Edificio Orión, 14014 Córdoba, Spain; fi.garciamaceira@gmail.com (F.I.G.-M.); titoanafe@gmail.com (L.A.T.-S.); nelebana83@hotmail.com (A.B.A.-G.); pazrojaselier@gmail.com (E.P.-R.); 2Department of Cell Biology, Physiology and Immunology, Campus de Excelencia Internacional Agroalimentario (ceiA3), University of Córdoba, 14014 Córdoba, Spain; bc1gorej@uco.es; 3Department of Physical Chemistry, Faculty of Pharmacy, University of Granada, 18071 Granada, Spain; rubino@ugr.es

**Keywords:** ELISA, antibodies immobilization, tetrazine, TCO, CEA

## Abstract

Enzyme-linked immunosorbent assay (ELISA) is routinely used to detect biomolecules related to several diseases facilitating diagnosis and monitoring of these, as well as the possibility of decreasing their mortality rate. Several methods have been carried out to improve the ELISA sensitivity through antibodies immobilization on the microtiter plates. Here, we have developed a strategy of antibodies immobilization to improve the ELISA sensitivity increasing the antibody density surface through the tetrazine (Tz)-trans-cyclooctene (TCO) reaction. For this, we prepared surfaces with tetrazine groups while the captured antibody was conjugated with TCO. The tetrazine surfaces were prepared in two different ways: (1) from aminated plates and (2) from Tz-BSA-coated plates. The surfaces were evaluated using two sandwich ELISA models, one of them using the low-affinity antibody anti-c-myc as a capture antibody to detect the c-myc-GST-IL8h recombinant protein, and the other one to detect the carcinoembryonic human protein (CEA). The sensitivity increased in both surfaces treated with tetrazine in comparison with the standard unmodified surface. The c-myc-GST-IL8h detection was around 10-fold more sensible on both tetrazine surfaces, while CEA ELISA detection increased 12-fold on surfaces coated with Tz-BSA. In conclusion, we show that it is possible to improve the ELISA sensitivity using this immobilization system, where capture antibodies bond covalently to surfaces.

## 1. Introduction

The enzyme-linked immunosorbent assay (ELISA) is one of the most used techniques for the detection of molecules, generally proteins, in different fields such as food technology, clinical diagnosis, scientific research, and industry. The sensitivity of an ELISA depends on different factors such as the interaction antigen-antibody, the temperature, the pH, and the effectiveness of the antibody binding to the test surface. Microtiter commercial plates use hydrophobic polymers such as polystyrene. The increase in hydrophilicity can enhance antibody binding (density) and decreases the amount of denatured protein. Therefore, the immobilization of the antibody on the polystyrene surface plays a crucial role in improving the detection limit [1].

It has been shown that a higher density of antibodies on the surface increases the detection sensitivity of analytes in ELISA, and, accordingly, different strategies have been developed to adequately immobilize proteins to surfaces [2,3,4]. These strategies are based on the generation of amino, carboxyl, or epoxy groups on the surface by different methods and the subsequent covalent bond of the antibody to these groups through chemical reactions using NH_2_, SH, and COOH free groups of the antibodies [5].

With the aim of increasing the sensitivity of the ELISA, Dixit et al. [6] used a methodology based on the use of aminopropyltriethoxysilane (APTES) to generate NH_2_ groups on the assay surface [4,5]. Subsequently, other groups followed this method with some variations (see, for example, the work of [7]). Using this approach, a 50-fold increase in sensitivity has been achieved compared to standard surfaces [8]. The capture of the antibody to the surface is carried out using EDC (carbodiimide) and NHS (N-hydroxysuccinimide) chemistry on the aminated 96-well plate, forming an amide-type covalent bond [5,9].

Additionally, other methods have been applied to produce functional groups on the surface, such as plasma treatment [10], corona discharge [11], ultraviolet treatment, and wet chemistry techniques [12]. The wet chemical modification allows chemical reactions to take place between a compound dissolved in an organic solution and the surface of the polymer, generating reactive functional groups on the surface [13]. This procedure shows the advantage that the treatment is penetrating and provides controlled and stable surfaces in terms of the number of functional groups and required typology, maintaining transparency in the case of ELISA plates [14]. Using chlorosulfonation-sulfonamidation reactions, aminated surfaces have been prepared as well, controlling the density of NH_2_ groups on the surface, which were used for the development of ELISAs [14].

Among the “click chemistry reactions”, which are frequently used in biomedicine, there is the “Inverse Electron Demand Diels–Alder” (IEDDA) reaction between 1,2,4,5-tetrazines (Tz) and trans-cyclooctenes (TCO), forming a dihydropyridazine bond in the absence of a catalyst [15]. Today, this one is among the fastest known click reactions, with second-order rate constants of up to 3.3 × 10^6^ M^−1^ s^−1^ [16]. The Tz-TCO reaction has been widely used in in vivo imaging studies, where the antibodies are labeled with TCO and later with tetrazine-technetium-99m [17,18], as well as for the immobilization of antibodies to surfaces [19].

In an attempt to increase the sensitivity of the ELISA technique, we used the click chemistry reaction between Tz-TCO, in which the capture antibody was modified with TCO, and the surface had exposed tetrazine groups. The addition of tetrazine to the plate was carried out in two different ways: (a) starting from aminated plates by wet chemistry and (b) coating the plate with the Tz-BSA protein by adsorption. To evaluate the sensitivity on both surfaces, we used a sandwich-type ELISA to detect the recombinant protein IL8h and the colorectal cancer biomarker CEA and compared the results obtained in the conventional ELISA with those achieved in the treated plates. The results show a significant increase in the sensitivity of the model ELISA for the detection of IL8h with respect to the conventional ELISA on both surfaces, as well as on the plates prepared with Tz-BSA in the case of the detection of CEA.

## 2. Materials and Methods

### 2.1. TCO Functionalization of Anti-c-myc and Anti-CEA

For both anti-CEA 3C1 (Hytest 4CA30) and anti-c-myc 9E10, TCO functionalization of antibodies was performed with UV-Tracer Trans-Cyclooctene NHS ester (Click Chemistry Tools, A1031) in phosphate buffer (0.15 M NaCl, 0.1 M NaPO_4_, pH = 7.5). Different molar relations between trans-cyclooctene and the antibody anti-c-myc were used to provide different degrees of labeling per antibody. Thus, 5, 10, and 15 equivalents of TCO per 1 equivalent of mAb were added. The antibody anti-CEA 3C1 was modified with 5 equivalents of TCO per 1 equivalent of antibody. Reactions were performed for 60 min at room temperature (RT) with stirring. TCO-modified antibodies were purified on Zeba desalting columns (40 kDa MW cut-off, 0.5 mL) (Pierce ZebaTM desalting columns, Thermo Scientific, Waltham, MA, USA).

The degree of labeling (DOL) was calculated based on the molar concentration relation between antibodies and reactive. Absorbance at 280 nm (A_280_) was used to determine the antibody concentration in the samples. Since UV tracer also absorbs at 280 nm, a correction factor was applied to adjust the A_280_ contributed by the tracer (A_280_ corrected = A_280_ − A_350nm_ × 0.4475). Then we checked the absorbance of TCO-modified antibodies at 280 nm and 350 nm, where antibodies molar extinction coefficient at 280 nm (ε) is equal ε_280nm_ = 204.000 M^−1^ cm^−1^ and UV tracer at 350 nm is equal at ε_350nm_ = 19.500 M^−1^ cm^−1^.

The affinity of the anti-c-myc-TCO antibodies conjugated toward their targets was assessed by ELISA and compared with the non-modified antibody. A microplate coated with 2 μg/mL c-myc-GST-human IL8 antigen was blocked with PBS containing 1% (*w*/*v*) BSA. Then, serially dilutions of the TCO-modified anti-c-myc antibodies were added and incubated at 37 °C for 1 h. Antibodies concentrations ranged from 0.39 to 50 ng/mL. Anti-mouse HRP (Sigma, A2554, Madrid, Spain) was used as a revealing antibody. Then ELISA colorimetric reaction employed TMB (3,3′,5,5′-tetramethylbenzidine), and it was stopped with 1 M HCl. The measure was performed at 450 nm. Similarly, the anti-CEA-TCO antibody affinity was measured by ELISA on a microplate coated with 5 µg/mL carcinoembryonic antigen (CEA) (Hytest, 8CEA88). TCO-modified anti-CEA 3C1 and unmodified antibody affinity were compared at concentrations ranging from 15.62 to 500 ng/mL.

### 2.2. Tetrazine Functionalization of BSA

Tetrazine functionalization of BSA was realized with methyl tetrazine–PEG4-NHS-ester (Click Chemistry Tools, 1069) using 30 equivalents of tetrazine per 1 equivalent of BSA in phosphate buffer (50 mM NaPO_4_, 150 mM NaCl, pH = 7). The reaction was developed under stirring in the dark at RT for 2 h and stopped with 1 M Tris-HCl buffer pH = 8. The reaction was dialyzed with 1 L PBS pH 7.4 using a dialysis tubing membrane of 12 kDa molecular weight cut-off. Tz-BSA concentration was determined by Bradford assay (Bio-Rad) at 495 nm and using bovine serum albumin (BSA) as standard DOL was calculated based on molar concentration relation between BSA and the reactive, considering that MW BSA = 66.433 g/mol and concentration of tetrazine was interpolated at the curve of reactive performed at A_520_.

### 2.3. Preparation of Microtiter Plates with Tetrazine

Aminated coated plates were prepared using a wet chemical technique by TECNALIA Laboratories (Project 2012/000069, TECNALIA Research & Innovation). The densities of NH_2_ groups were 1.5 and 3 nmol/cm^2^. Microtiters well were treated with 5 equivalents of methyl tetrazine–PEG4-NHS-ester per NH_2_ group in 100 µL of phosphate buffer (50 mM NaPO_4_, 150 mM, NaCl, pH = 7). Plates were stirred in the dark for 2 h at RT following by 3 washing steps with 0.3 mL of PBS pH = 7.4.

Standard microplates of polystyrene (MaxiSorp, Thermo 442404, Waltham, MA, USA) were coated with 20, 50, or 100 μg/mL of Tz-BSA in buffer 0.1 M NaHCO_3_ pH = 9 and incubated for 16 h at 4 °C. After 5 washing steps with 0.3 mL of PBS-Tween-20 0.05%, plates were treated with coating stabilizer and blocking buffer (Sigma-Aldrich, C9483) and stored at 4 °C.

### 2.4. ELISA to Evaluate Binding of TCO-Modified Antibodies to Tetrazine Surfaces

Standard microplates coated with 20 µg/mL of Tz-BSA or aminated surface treated with tetrazine were added serially dilutions of the TCO-modified anti-c-myc antibody or the unmodified antibody and incubated at 37 °C for 1 h. Antibodies concentration ranged from 15.6 to 2000 ng/mL. After microplate washing, anti-mouse HRP and TMB were sequentially added. The enzyme reaction was stopped with 1  M HCl, and color development was measured at 450  nm on a microtiter reader.

### 2.5. ELISA Assays for the Detection of Bi-Specific Model c-myc-GST-IL8h Protein Using Treated Tetrazine Surfaces

The sandwich ELISA was used for the detection of human c-myc-GST-IL8h as well as to evaluate the tetrazine surfaces and to compare the limits of detection obtained in this surface with the results achieved from standard surfaces. We compared 1.5 nm/cm^2^ and 3 nm/cm^2^ of NH_2_ group in tetrazine surface prepared from wet chemical and the concentration of the coating with Tz-BSA, 20, 50, or 100 µg/mL on standard plates.

The ELISA was performed as described by García-Maceira et al. [20] with modifications. Briefly, standard plates coated with Tz-BSA or aminated plates Tz treated were added 2 μg/mL anti-c-myc-TCO in 0.1 M Na_2_CO_3_/NaHCO_3_ pH =  9.6. Simultaneously standard plates were coated with unmodified anti-c-myc antibody and blocked with PBS pH = 7 containing 1% (*w*/*v*) BSA. The following ELISA steps were executed as described below. All plates were added to the model protein c-myc-GST-IL8h in serial dilutions from 0.78 to 100 ng/mL. ELISAs were then developed using an anti-human IL-8 biotin clone MT8F19 antibody at 1 μg/mL. After 1 h of incubation at 37 °C and 5 washing steps, streptavidin-HRP (Thermo Fisher, 21130, Waltham, MA, USA) diluted at 100 ng/mL was added. ELISA readout was as described previously.

### 2.6. Comparative ELISA Assays for the Detection of Carcinoembryonic Human Protein Using Treated Tetrazine Surfaces

The standard and Tz-BSA-coated surfaces were compared using the CEA sandwich ELISA. MaxiSorp plates were coated with 5 μg/mL unmodified anti-CEA 3C1 antibody in 0.1 M Na_2_CO_3_/NaHCO_3_ pH = 9.6 and blocked with PBS pH = 7 containing 1% BSA. Simultaneously, anti-CEA-TCO was added to 20 μg/mL of Tz-BSA pre-coated plates. The next steps were the same for both ELISA. A range of concentrations of CEA protein (0–200 ng/mL) was added per duplicate in wells and incubated for 1 h at RT. After 3 washing steps, detection was achieved with 100 µL of 0.5 μg/mL of biotin anti-CEA 3C6. After 5 washing steps, 100  μL per well of streptavidin-HRP was diluted at 100  ng/mL, and TMB was sequentially added. The enzyme reaction was stopped with 1 M HCl, and color development was measured at 450 nm on a microplate reader.

### 2.7. Assay Performance Analysis

The results of ELISAs were analyzed using the GraphPad Prism program (GraphPad Software Inc., San Diego, CA, USA), and the four-parameter logistic standard curve showed the EC50 and R^2^ of each one. The detection limit was calculated using the absorbance arithmetic media of all repetitions of the blank plus 3 standard deviations. Similarly, was calculated the limit of quantification using 10 standard deviations as criteria. Finally, the LOD and LOQ concentrations were obtained by plotting the absorbance of each other on the standard curve.

## 3. Results

Coating ELISA antibodies were modified to include TCO functional groups through their NH_2_ groups. This modification was not directional, and TCO groups would be attached at any NH_2_ sites of the molecule. Modifications of anti-c-myc 9E10 with 5, 10, and 15 equivalents of TCO per antibody molecule resulted in anti-c-myc-TCO conjugates with 4, 8, and 10 TCOs/antibody, respectively. On the other hand, anti-CEA 3C1 antibody modified with five equivalents of TCO resulted in anti-CEA-TCO conjugates with four TCOs/molecule.

The affinity of anti-c-myc-TCO-conjugated antibodies was compared with anti-c-myc unmodified through an indirect ELISA coated with c-myc-GST-IL8h protein and revealed with anti-mouse IgG-HRP. As observed in Figure 1, this modification did not disturb the affinity toward c-myc when the antibody contains four TCO groups per molecule as the curve was like that one obtained with the unmodified antibody, being EC50 values also similar. In contrast, with 8 and 10 TCO groups per antibody, the specific antigen anti-c-myc detection activity was significantly reduced. Consequently, we selected the conjugation process with five equivalents.

Similarly, using an indirect ELISA, we confirmed that anti-CEA-TCO conserved the same activity that the unmodified antibody.

### 3.1. Evaluation of Tetrazine-Coated Microtiter Plates

The Tz-BSA protein was assessed at 520 nm to quantify the number of tetrazine groups incorporated into a BSA molecule. The protein concentration was measured by Bradford using a standard curve of unmodified BSA. Considering an MW of BSA as 66.433 gmol^−1^, we calculated the molar concentration of Tz-BSA, and the absorbance at 520 nm was interpolated at a linear regression curve of methyl tetrazine–PEG4-NHS-ester. Finally, we obtained a ratio of 25 tetrazines per molecule of BSA.

The evaluation of the binding to tetrazine-coated surfaces was performed with the anti-c-myc-TCO (1:4) antibody. As we show here, the anti-c-myc-TCO antibody bound specifically to the tetrazine surfaces, either pre-coated with Tz-BSA or aminated plates treated with tetrazine, and binding of the anti-c-myc antibody lacking TCO conjugation was not observed (Figure 2). There were no differences between both different tetrazine prepared surfaces.

The incubation time of the click chemistry reaction between tetrazine surface and TCO from the anti-c-myc-TCO conjugated antibody was measured at 60, 120, 180, 240, and 300 min with concentrations of antibody of 100, 200, 500, 1000, 2000, and 5000 ng/mL. This reaction was performed on Tz-BSA pre-coated surface. As observed in Figure 3, the absorbance did not increase further after 2 h incubation, remaining unchanged independently of the antibody concentration. Nevertheless, there was a signal increase when antibody concentration was raised. The optimal results were obtained when using more than 1000 ng/mL (see Figure 3).

### 3.2. ELISA Assays for the Detection of c-myc-GST-IL8h Protein Using Treated Tetrazine Surfaces

The recombinant protein c-myc-GST-IL8h was cloned and expressed by Canvax Biotech S.L. García-Maceira et al. [20] confirmed that this protein can be detected by ELISA with different capture antibodies, as anti-human IL8 or anti-c-myc 9E10 antibody, using the same revealed biotin anti-IL8h antibody, so that it can be used as a chimeric protein and would be detected by the c-myc tag or IL8h protein [20]. Significative differences in the detection of the recombinant protein with different coating antibodies were also found [20], indicating that a low-affinity antibody reduces the assay sensitivity. Accordingly, we designed a strategy to improve an ELISA developed with an antibody that can decrease this effect.

Tetrazine surfaces were prepared in two different ways. Polystyrene plates were treated by wet chemical to the prepared aminated surfaces, followed by a reaction with NHS-tetrazine reagent obtaining tetrazine surfaces. The aminated surfaces used contained 1.5 and 3 nmol/cm^2^ of NH_2_ groups per well. The second method consisted of Tz-BSA pre-coated standard surface.

The detection sandwich ELISA of c-myc-GST-IL8h was performed to determine the best concentration of NH_2_ groups on surfaces prepared by the wet chemical technique. These surfaces were the starting point to prepare tetrazine surfaces. We compared 1.5 and 3 nmol/cm^2^ of NH_2_ groups per well using a standard plate as control (Figure 4). LOD and LOQ were calculated from 4PL fitted curves, and significant differences between the ELISA performed on tetrazine surface with respect to standard were found. Thus, the sensitivity on tetrazine surface prepared from aminated wells with 1.5 nmol/cm^2^ was 5-fold higher than on standard surfaces, while tetrazine surfaces prepared from 3 nmol/cm^2^ of NH_2_ were 11-fold more sensitive than the standard control (Table 1). Although there were no visual differences between both tetrazine surface curves, LOD and LOQ values showed that the ELISA developed on tetrazine surface prepared from 3 nmol/cm^2^ of NH_2_ was more sensitive than the one developed on a surface prepared from 1.5 nmol/cm^2^ of NH_2_. These differences can be attributed to the standard deviation in tetrazine surface prepared from 1.5 nmol/cm^2^ of NH_2_ since it was higher than the one prepared from 3 nmol/cm^2^.

On the other hand, we coated standard ELISA (MaxiSorp) surfaces with different concentrations of Tz-BSA (25 tetrazine: 1BSA) protein (20, 50, and 100 µg/mL) to develop the ELISA for c-myc-GST-IL8h detection. The capture antibody used was anti-c-myc-TCO (1:4), and the revealing antibody was anti-human IL8 biotin. As depicted in Figure 5, no differences in the ELISAs sensitivity were detected, as LOD and LOQ results were similar. Considering these results and using the statistical concept of medium square, we calculated *p*-value for LOD = 0.015 ng/mL and for LOQ = 0.006 ng/mL with a 95% probability. These results revealed that there were no significant differences between the three concentrations of Tz-BSA to coat and, therefore, we selected a 20 µg/mL concentration of Tz-BSA for the next assays.

The sandwich ELISA for c-myc-GST-IL8h detection was repeated three times to compare the LOD and LOQ in tetrazine and standard surfaces. In Table 2, we show the results of each assay as well as the standard deviation and coefficient of variation inter-assay. In the standard surface, the medium LOD was 11.17 ± 0.62 ng/mL and the LOQ was 14.4 ± 0.82 ng/mL, while in the tetrazine surface prepared from pre-aminated plates, these values were 0.96 ± 0.06 ng/mL and 1.34 ± 0.14 ng/mL, respectively. On another tetrazine surface prepared by coating the standard surface with Tz-BSA, LOD and LOQ values were 0.77 ± 0.05 ng/mL and 1.01 ± 0.14 ng/mL, respectively. Concerning the assays sensitivity, we found an improvement in both tetrazine surfaces with respect to the standard. Thus, in tetrazine surfaces prepared from pre-aminated plates, the assay was approximately 11-fold more sensitive, while in pre-coated Tz-BSA surfaces, it was 14-fold compared to the standard surface. The significant differences between groups shown in Figure 6 confirm that both tetrazine surfaces improve the detection sensitivity of ELISA of c-myc-GST-IL8h.

Significant differences were found when comparing both tetrazine prepared surfaces (Figure 6). Since the preparation of Tz-BSA-coated surface is a simple process compared with tetrazine surfaces prepared from aminated plates, we selected this surface in another ELISA comparative assay that includes an important antigen with clinical interest.

### 3.3. ELISA Assays for the Detection of CEA Using Tetrazine-Treated Surfaces

We developed another sandwich ELISA using Tz-BSA pre-coated surfaces to explore possible sensitivity improvement of ELISA through the tetrazine-TCO reaction compared with the standard surface, in which the antibody binds to the surface by passive adsorption. When comparing the signals observed in the Tz-BSA surface and those obtained in the standard one, we detected a higher signal in TZ-BSA with the same concentrations of CEA and with a clear displacement of the curve to the left compared with the standard curve (Figure 7A). This ELISA comparison was developed three times, and the LOD in the standard surface was 7.47 ± 0.41 ng/mL, while in Tz-BSA-coated surface, it was 0.60 ± 0.06 ng/mL. There was an increased sensitivity in CEA detection ELISA of 12.4-fold compared with the traditional technique. A Student t-test showed extremely significant differences between both surfaces (*p* < 0.0001; see Figure 7B), which allows us to conclude that Tz-BSA-coated plates combined with modified-TCO capture antibody greatly improve the ELISA sensitivity.

## 4. Discussion

There are significant differences in protein detection sensitivity by ELISA depending on the capture antibody. This fact was reported by García-Maceira et al. [20], who developed an ELISA with different capture antibodies to detect a chimeric recombinant protein c-myc-GST-IL8h, showing differences in the dynamic range of concentrations and the sensitivity for the analyte detection [20].

Considering the model of Langmuir [21], the detection limit of an ELISA depends on the binding affinity of the capture antibody with the target protein. If antibody-antigen binding constant values increase two orders of magnitude, the detection limit values decrease proportionally [22]. In agreement with most of the reports, binding constant values may vary in a wide range: from 10^−5^ to 10^−12^M^−1^ [23]. Since there are many antibodies with low affinity, several strategies have been recently developed to minimize their effect on ELISA sensitivity.

Different crosslinkers have been employed to immobilize proteins covalently on aminated polystyrene surfaces, such as glutaraldehyde, with each extreme able to bind to an NH_2_ group, or carbodiimide, which links an amino group with a carboxyl [8,24]. Surfaces activated with APTES, with NH_2_ free groups on the surface, have been used for the covalent immobilization of antibodies [5,7,24]. Dixit et al. [5] used a combination of the cross-linker carbodiimide/NHS to improve an ELISA of fetuin-A detection, improving the sensitivity 16-fold compared to commercial kits. On the other hand, Wang et al. [7] used glutaraldehyde as a crosslinking to immobilize HIV-p24 antigen on surfaces activated with APTES, improving by 30-fold the LOD compared with the standard surface in an ELISA of HIV-p24 detection [7].

In this work, we did not use crosslinkers such as glutaraldehyde or carbodiimide because when used in situ, they could form bonds between the antibody molecules causing its precipitation and loss of activity. The activity evaluation of the modified antibody before being used offers several advantages: (a) to decrease the inter-assay variability and thus increasing its reproducibility; (b) to store the modified antibody until used, being then available each time the ELISA is repeated, and (c) it could even be marketed once it had been modified. In this work, we prepared capture antibodies pre-modified with TCO and plates prepared with tetrazine on the surface, and both were pre-evaluated and stored before their use so that when they are mixed, a covalent bond is formed between tetrazine and TCO groups with high specificity and selectivity.

Several bioorthogonal reactions, including the azide-alkyne [3+2] cycloaddition, the photoinducible 1,3-dipolar cycloaddition, and the inverse electron demand Diels–Alder (IEDDA) reaction, have been developed and used for several applications, including protein modifications [25,26]. Among these, the IEDDA reaction stands out from other bioorthogonal reactions due to its unmatchable kinetics (rate constant of up to 10^6^ M^−1^ s^−1^) and its high specificity and biocompatibility [25]. Specifically, this reaction has been used to conjugate antibodies, which were later conjugated with tetrazine or TCO and then applied to cell imaging, nanoparticle functionalization, and targeted antibody therapy [27,28]. Attending to its characteristics and applications, we selected this reaction as the basis of system development to increase ELISA sensitivity.

The capture antibodies used in this work were modified with TCO through their free NH_2_ groups using NHS reagent. We observed that when the modification compound increased with respect to the antibodies, the TCO groups incorporated into the antibody molecules increased too. However, decreased antibodies activity was also detected, demonstrating the importance of the evaluation of the modified antibodies.

On the other hand, the microtiter ELISA plates were prepared with tetrazine in two different ways: from aminated plates after wet chemical treatment of chlorosulfonation-sulfonamidation and from MaxiSorp plates coated with the BSA protein modified with 20 tetrazines per protein molecule. Aminated surface prepared using wet chemical technique has some advantages compared with other methods described of polystyrene preparation surface with reactive groups [29]. Thus, in this method, polystyrene surface is activated with chlorosulfonyl groups, and then a bifunctional aliphatic amine compound is added, allowing an amine group linkage to the surface, while the second functional group remains available for further reactions. This method also allows to acquire a controllable number of amine groups on polystyrene surfaces and additionally provides a higher amount of amine groups than that obtained using commercial plasma, functionalized polystyrene substrates, or ultraviolet treatment [13].

Attending the results obtained for LOD and LOQ calculated for c-myc-GST-IL8h detection in the different surfaces (see Table 2), we found an improvement of the sensitivity in both tetrazine surfaces when an anti-c-myc-TCO was used as a capture antibody compared with the standard surface. Thus, ELISA sensitivity increased in one order of magnitude. In this sense, when using Tz-BSA surfaces, the LOD was 13.7-fold improved compared to the standard plates, while in tetrazine surfaces followed by wet chemical treatment, this improvement was 11-fold. On the other hand, the surfaces prepared with Tz-BSA showed increased sensitivity for the detection of CEA, an effect due to a 12-fold LOD improvement when compared with ELISAs developed by traditional techniques. These results demonstrate that the system using Tz-TCO reaction on the surface of ELISA plates for antibody anchoring significantly improves the sensitivity of the assay and could become a routine method in research laboratories due to its reduced complexity.

The Tz-BSA-coated surface preparation is less laborious than the tetrazine prepared plates after wet chemical amination. In addition, the Tz-BSA protein can be stored, and the surfaces can even be previously coated with the protein by using a stabilizer reagent so that, like the TCO-labeled antibody, the surface would be readily available when needed, and eventually, it could also be marketed.

## 5. Conclusions

In this work, we describe a method based on Tz-TCO reaction to immobilize antibodies to microtiter surfaces that additionally may improve the ELISA assay sensitivity, using for this purpose the comparison of sandwiches ELISA with their corresponding traditional methods. The Tz-BSA-coated surface also provides a simple procedure to obtain surfaces with a high quantity of tetrazine that increases ELISA sensitivity.

## Figures and Tables

**Figure 1 biosensors-11-00524-f001:**
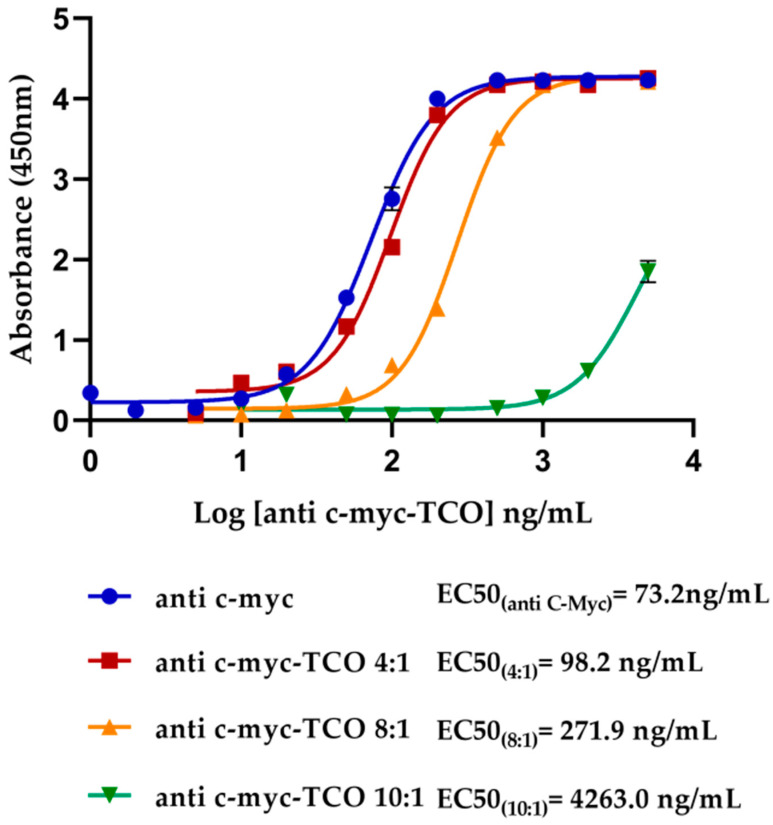
Comparative ELISA of anti-c-myc-TCO-conjugated antibodies and anti-c-myc unmodified to its antigen affinity c-myc. A 4PL sigmoidal curve to each anti-c-myc antibody is displayed. The EC50 represents the analyte concentration capable of generating half of the maximum signal obtained in the test in the linear range of the curve. The error bars correspond to three replicates of each antibody concentration.

**Figure 2 biosensors-11-00524-f002:**
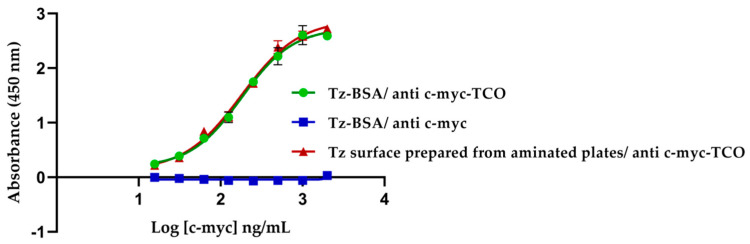
Evaluation of tetrazine surfaces using anti-c-myc-TCO (4:1). The 4PL sigmoidal curves of antibodies concentration against absorbance at 450 nm. The negative control was performed on Tz-BSA-coated surface. The error bars correspond to three replicates of each concentration of antibody.

**Figure 3 biosensors-11-00524-f003:**
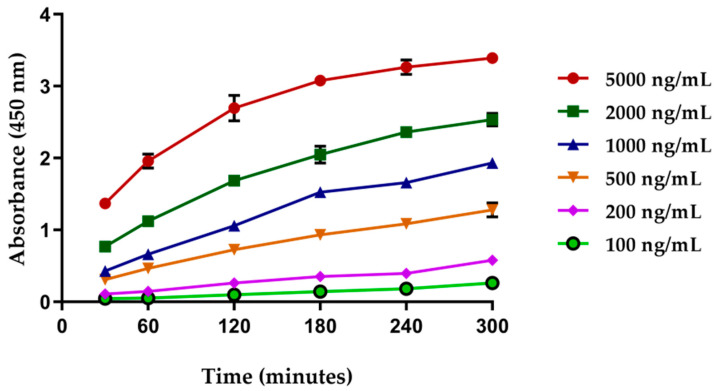
Incubation time optimization of the tetrazine-TCO reactions on the surface of microtiter plates prepared with tetrazine using the anti-c-myc-TCO (1:4) antibody. Four replicates were carried out for each antibody concentration and incubation time. Unmodified anti-c-myc antibody on Tz-BSA pre-coated wells. Anti-c-myc-TCO antibodies on pre-coated BSA wells were used as negative controls. In these cases, signals were too low to be represented in the graph.

**Figure 4 biosensors-11-00524-f004:**
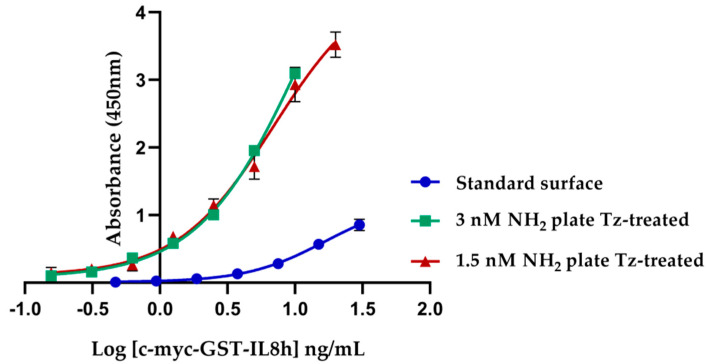
Comparative ELISA of c-myc-GST-IL8h detection in standard and tetrazine-coated surfaces. 4PL sigmoidal curves on two tetrazine surfaces prepared from aminated plates with different NH_2_ groups density and standard surface were used as control. On the tetrazine surface, the capture antibody was anti-c-myc-TCO, while on the standard surface, we used an anti c-myc unmodified antibody. The results correspond to three replicates for each concentration. The negative control included 10 replicates.

**Figure 5 biosensors-11-00524-f005:**
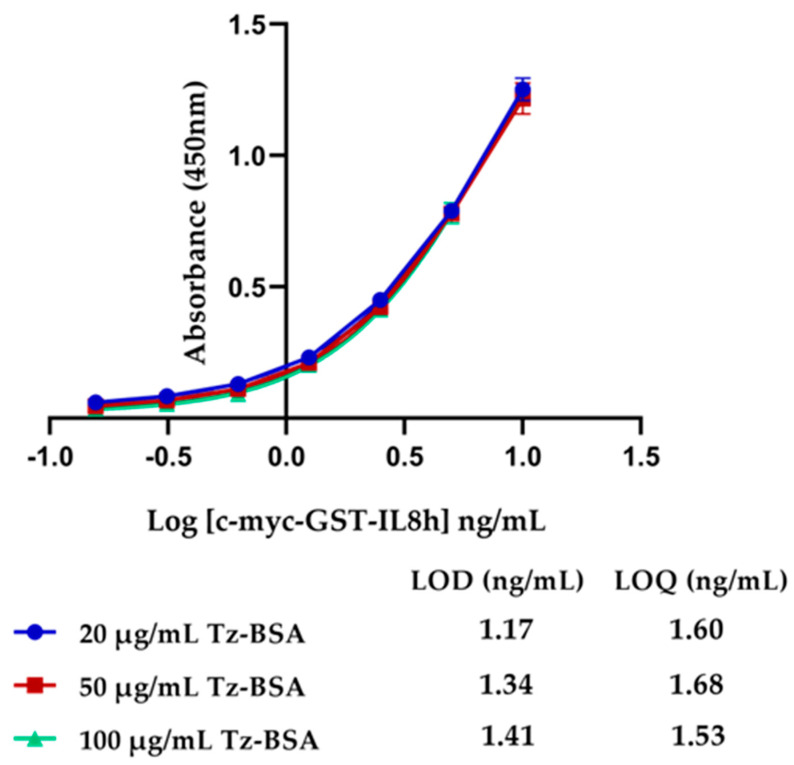
ELISA of c-myc-GST-IL8h detection in standard surfaces pre-coated with different concentrations of Tz-BSA. The 4PL sigmoidal curves to each coated Tz-BSA surface are represented.

**Figure 6 biosensors-11-00524-f006:**
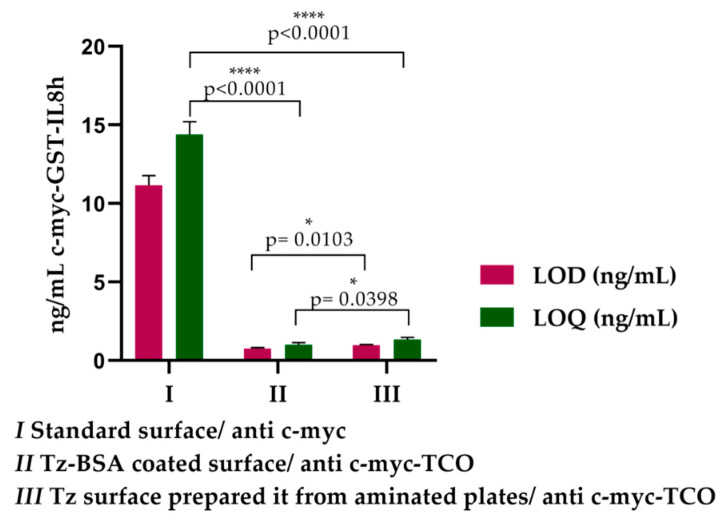
Bar graphs of sandwich ELISA for c-myc-GST-IL8h protein detection representing the arithmetic media of LOD and LOQ (ng/mL) calculated for Tz-BSA-coated surface, tetrazine surface prepared from pre-aminates plates, and standard surface.

**Figure 7 biosensors-11-00524-f007:**
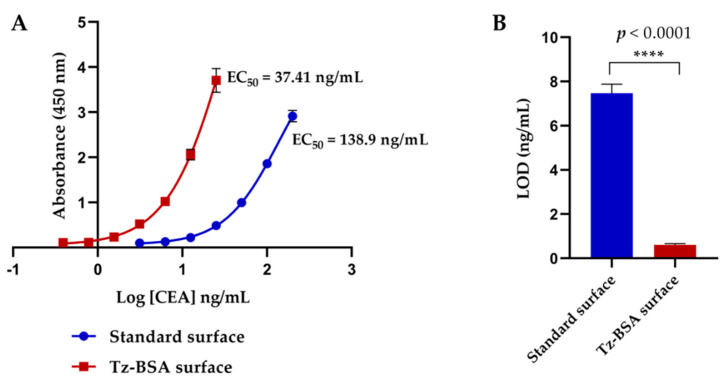
ELISA for CEA protein detection developed on Tz-BSA-coated and standard surfaces. (**A**) 4PL sigmoidal curves. (**B**) Bar graphs representing mean values of LOD (in ng/mL) calculated for Tz-BSA-coated and standard surfaces showing significant differences between both conditions (*p* < 0.0001).

**Table 1 biosensors-11-00524-t001:** Comparison of LOD and LOQ values obtained for tetrazine surfaces prepared from aminated plates with different concentrations of amine groups per well, with respect to the standard plates.

	Standard Surface	Tz Surface Prepared from Aminated Plates with	
		3 nmoles NH_2_/cm^2^	1.5 nmoles NH_2_/cm^2^
LOD (ng/mL)	10.90	0.81	1.97
LOQ (ng/mL)	13.83	1.22	2.74
R^2^	0.9997	0.9987	0.9948

**Table 2 biosensors-11-00524-t002:** LOD and LOQ values of ELISA detection of c-myc-GST-IL8h protein. The results of three different independent assays for each surface are shown.

Surfaces		LOD (ng/mL)	LOQ (ng/mL)
Standard Surface	Replicate 1	11.52	14.63
	Replicate 2	10.45	13.49
	Replicate 3	11.52	15.07
	Arithmetic media	11.17	14.40
	Standard deviation	0.62	0.82
	%CV inter-assay	5.54	5.66
Tz-BSA-Coated Surface	Replicate 1	0.70	0.91
	Replicate 2	0.79	1.16
	Replicate 3	0.80	0.95
	Arithmetic media	0.77	1.01
	Standard deviation	0.05	0.14
	%CV inter-assay	7.01	13.56
Tetrazine Surface Prepared from Aminated Plates	Replicate 1	0.92	1.37
	Replicate 2	0.92	1.19
	Replicate 3	1.03	1.46
	Arithmetic media	0.96	1.34
	Standard deviation	0.06	0.14
	%CV inter-assay	6.64	10.26

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
