# Peer review of "Covalent Immobilization of Antibodies through Tetrazine-TCO Reaction to Improve Sensitivity of ELISA Technique"

_biosensors, 2021, doi:10.3390/bios11120524_

Round 1
Reviewer 1 Report
This manuscript reported a strategy of antibodies immobilization to improve the ELISA sensitivity increasing the antibody density surface through the Tetrazine (Tz) - Trans-cyclooctene (TCO) reaction. For this purpose, they have been prepared surfaces with tetrazine groups, while capture antibody was conjugated with TCO. The sensitivity increased in the surfaces treated with tetrazine in comparison with the standard unmodified surface.
The work has appeared to be rigorously performed. The paper is overall clear, accurately layed out and well constructed.
In general, the results of this work are interesting and the conclusions are supported by the experimental details. Some minor points should be taken into consideration prior to publication:
- Did the authors investigate if the antibodies were released after immobilization process?
-
Check the reference based on journal guidelines.
Author Response
Dear Reviewer:
We are pleased with your comments about the investigation of the manuscript. We respond all your suggestions below.
(x) English language and style are fine/minor spell check required
R: The English version has been thoroughly revised. All the changes have been appropriately marked with the “track changes function”.
This manuscript reported a strategy of antibodies immobilization to improve the ELISA sensitivity increasing the antibody density surface through the Tetrazine (Tz) - Trans-cyclooctene (TCO) reaction. For this purpose, they have been prepared surfaces with tetrazine groups, while capture antibody was conjugated with TCO. The sensitivity increased in the surfaces treated with tetrazine in comparison with the standard unmodified surface.
The work has appeared to be rigorously performed. The paper is overall clear, accurately layed out and well-constructed.
In general, the results of this work are interesting, and the conclusions are supported by the experimental details. Some minor points should be taken into consideration prior to publication:
Did the authors investigate if the antibodies were released after immobilization process?
R: This is a very interesting issue as it would help to understand the mechanism behind the increased sensitivity of covalent immobilization vs physical adsorption. However, we did not performed assays to quantify the amount of antibody immobilized and the amount released.
The maximum number of antibodies that can be immobilized on the surface of the ELISA 96 well microplate is about 1.3x1012 molecules; that is, about 300 ng of antibodies are attached per well. Assuming that up to a 5% of bound antibodies could be released by washing with 200 µL, then the concentration of released antibodies is below 75 ng/mL when physical adsorption is used for antibody immobilization. When covalent immobilization is used, the expected amount that could be released is even lower and hardly could be quantified. Moreover, in our system we did not expect a loss of antibodies by washing because the bond between TCO and tetrazine is covalent and, therefore, it is difficult to release the antibody without using reagents for antibody fragmentation.
Check the reference based on journal guidelines.
R: We checked all references according to the journal guidelines
Thank you
Kind regaard
Tania García Maceira
Reviewer 2 Report
The authors report on a strategy of antiboties immobilization which improve ELISA sensitivity. In general, the manuscript is well organized and data support the conclusions. My major concern is that the focus is on immobilization, which is a relevant step for biosensor development but no biosensors devices are discussed here. Futhermore the manuscript would benefit from a moderate English revision (e.g. line 114) and subsections should be numbered.
Author Response
Dear Reviewer:
We are pleased with your revision of the manuscript. We answered all your suggestions below.
(x) Moderate English changes required
R: The English version has been thoroughly revised. All the changes have been appropriately marked with the “track changes function”
The authors report on a strategy of antibodies immobilization which improves ELISA sensitivity. In general, the manuscript is well organized, and data support the conclusions. My major concern is that the focus is on immobilization, which is a relevant step for biosensor development, but no biosensors devices are discussed here. Furthermore, the manuscript would benefit from a moderate English revision (e.g., line 114) and subsections should be numbered.
R: The English version of the manuscript has been carefully revised (see “track changes function”) and we have incorporated numbers to the subsections, which facilitates the reading of the paper. Regarding biosensor devices, our work is more related to “biosensing” and that is why we do not discuss about devices. Despite this, we believe that this work fits well within the scope of the journal and may be of great interest to Biosensor’s readers, as we describe an easy and feasible method to increase the sensitivity of ELISA assays.
Thank you very much
Kind regard
Tania García-Maceira